# Novel Stable Capacitive Electrocardiogram Measurement System

**DOI:** 10.3390/s21113668

**Published:** 2021-05-25

**Authors:** Chi-Chun Chen, Shu-Yu Lin, Wen-Ying Chang

**Affiliations:** 1Department of Electronic Engineering, National Chin-Yi University of Technology, Taichung 41170, Taiwan; steven870408@gmail.com; 2Department of Electronic Engineering, 3R Life Sciences Taiwan LTD, Kaohsiung 821, Taiwan

**Keywords:** noncontact electrocardiogram, surface guard ring, optimal input resistance, optimal voltage divider feedback

## Abstract

This study presents a noncontact electrocardiogram (ECG) measurement system to replace conventional ECG electrode pads during ECG measurement. The proposed noncontact electrode design comprises a surface guard ring, the optimal input resistance, a ground guard ring, and an optimal voltage divider feedback. The surface and ground guard rings are used to reduce environmental noise. The optimal input resistor mitigates distortion caused by the input bias current, and the optimal voltage divider feedback increases the gain. Simulated gain analysis was subsequently performed to determine the most suitable parameters for the design, and the system was combined with a capacitive driven right leg circuit to reduce common-mode interference. The present study simulated actual environments in which interference is present in capacitive ECG signal measurement. Both in the case of environmental interference and motion artifact interference, relative to capacitive ECG electrodes, the proposed electrodes measured ECG signals with greater stability. In terms of R–R intervals, the measured ECG signals exhibited a 98.6% similarity to ECGs measured using contact ECG systems. The proposed noncontact ECG measurement system based on capacitive sensing is applicable for use in everyday life.

## 1. Introduction

Electrocardiograms (ECGs) are the most commonly used bioelectric signal. ECGs are used for cardiovascular disease screening and assessing heart or cardiovascular functions [1,2,3,4,5,6]. Additionally, they have become increasingly critical in lifestyle and consumer applications, including exercise monitoring [7,8], fatigue detection [9,10], and stress monitoring [11,12]. Conventional ECG measurement methods generally use Ag/AgCl electrodes or dry electrodes to make direct contact with the human body, employ electronic devices to amplify and digitize signals, and perform wave processing to determine the ECG waveform. However, such conventional ECG recording methods have a severe drawback: the electrodes must be in contact with human skin. This contact is required for both short ECG recording using conductive gel electrodes and long ECG recording using dry electrodes. In practice, this requirement limits the convenience and applicability of ECG measurement.

Recent studies have proposed and investigated noncontact, capacitive ECG (CECG) electrodes [13,14,15,16,17,18,19], which mainly employ capacitive coupling between surface electrodes to measure biological signals. By using the human skin and the electrode as two ends of a capacitor, capacitive electrodes can transmit the bioelectrical signal through the capacitor. Capacitance electrodes were first proposed by Richardson [20]. However, the shortcoming of capacitive electrodes is that the signal quality is much lower than that of contact ECG. A high input impedance amplifier is embedded in the electrode to overcome this shortcoming [21,22,23]. In the following decades, the design and performance of capacitive electrodes have continued to improve to obtain high-quality ECG signals [24,25,26]. The emergence of noncontact electrodes has increased the popularity of measurement applications. For example, Lim et al. embedded capacitive electrodes inside mattresses [27], Baek et al. applied capacitive electrodes to chairs [28], and Nemati et al. embedded CECG electrodes in T-shirts [29]. One study installed capacitive electrodes in car seats to measure the ECG of drivers [30]. However, CECG electrode signals are susceptible to interference, which prevents them from accurately obtaining the waveform. Therefore, measuring ECG signals in a stable and accurate manner is pivotal to increasing the feasibility of CECG electrode applications.

The three main interference sources for CECG measurement are common-mode noise generated by power cords [31], motion artifacts [32], and interference from the external environment [33]. Such interference may cause signal distortion and saturation when measuring physiological signals, minimizing their accuracy. Thus, various noise reduction methods have been proposed. For example, Lee et al. employed conductive foam to reduce motion artifact interference generated by the friction between electrodes and clothing [34]. Serteyn et al. adopted an injection signal to overcome common-mode noise [35], and Eilebrecht et al. applied an additional accelerator sensor to offset motion artifact interference [36]. Other methods for overcoming common-mode noise are also available, for example, driven right leg (DRL) [37] and capacitive DRL (CDRL) [38]. Despite various methods having been proposed to overcome interference when measuring with CECG electrodes, to the best of the author’s knowledge, few methods have achieved stable CECG signal measurement in real-life settings.

Therefore, the present study proposed a novel noncontact ECG circuit measurement system to address this limitation. The system employs a high gain circuit design to increase the signal-to-noise ratio (SNR). For example, use of the optimal input resistance can optimally reduce the effects of the bias current; the guard ring reduces electromagnetic wave interference from the external environment, and the output resistance divider feedback reduces the influence of parasitic capacitance. This robust design enables the circuit to stably measure the CECG signal regardless of external or motion artifact interference caused by the circuit. Furthermore, the circuit is simple and practical. The proposed circuit may be applied in noncontact ECG measurements to increase their feasibility.

## 2. Materials and Methods

### 2.1. Active Electrode Design

The electrode design prioritizes collecting signals with a relatively high SNR. In addition to including a guard ring to shield the electrodes from external noise, the design employs high input impedance (*R_load_*) to increase the gain and the output resistance divider feedback circuit to reduce the influence of stray capacitance. Throughout the design process, the effect of passive components on the electrode circuit was adjusted to obtain the optimal component combination. Figure 1a,b presents the equivalent circuit and photos of the electrode, respectively. The diameter of the electrode is 2.5 cm. Given that the operational amplifier (OPA) has infinite input impedance, the output impedance can be disregarded. The circuit gain equation is presented in Equation (1), where Z_I_ denotes the impedance of a combined component, where the stray capacitance (*C_stray_*) is first connected in series to the ground guard ring capacitor (*C_gnd_*) and then connected in parallel to the *R_load_*, and *Z_Ce_* is the capacitive impedance of clothing. The measured value of *C_e_* is 47 pF, *C_stray_* is 76 pF, and *C_gnd_* is 31 pF.
(1)$$G\left(s\right)=\frac{Z_{I}}{Z_{Ce}+Z_{I}}$$

The system employs a resistance divider feedback circuit to reduce the effect of stray capacitors. The effects of the stray capacitor (${C^{\prime }}_{stray}$) are calculated using Equation (2).
(2)$${C^{\prime }}_{stray}=\frac{R_{fl}}{R_{fh}+R_{fl}}C_{stray}$$

Accordingly, the gain is calculated using Equation (3).
(3)$$G\left(s\right)=\frac{sC_{e}R_{load}\left({C^{\prime }}_{stray}+C_{gnd}\right)}{{C^{\prime }}_{stray}+C_{gnd}+sR_{load}\left({C^{\prime }}_{stray}C_{gnd}+C_{e}C_{gnd}+C_{e}{C^{\prime }}_{stray}\right)}$$

#### 2.1.1. High-Input Impedance Amplifier

A high-input impedance OPA is necessary for electrodes to obtain ECG signals through clothing. The LMP7721 (TI, Dallas, TX, USA) was adopted as the head amplifier; its input impedance and capacitance are >1 TΩ and <2 pF, respectively. Other outstanding functions of this OPA include a high common-mode rejection ratio (100 dB), low input voltage noise (6.5 nV/√Hz), low input bias current (3 fA), and the functions that are used to reduce noise and offsets across the high source and bias impedance. The OPA achieves low voltage (1.8–5.5 V) and output (1.3 mA) operation, thereby making it feasible for portable battery supply systems.

#### 2.1.2. Input Resistor

A simple and robust direct current bias architecture was implemented by adding a resistor from input to ground. The input resistor (*R_load_*) provides a stable bias current path for the preamplifier. To simulate the high impedance of clothing, studies have employed several gigaohms of resistance for matching [34,39]. To determine the optimal resistance value, the simulated results of resistance values between 100 MΩ and 100 GΩ were compared. In Figure 2a, the resistance value determines the cutoff frequency of the high-pass filter, which is a combination of the capacitive coupling of clothing and the resistor. A resistance value of <1 GΩ is unsuitable for ECG measurement because it would attenuate the ECG signal. The main bandwidth of ECG signals is 0.5–35 Hz. Commercial power sources, the largest source of electrode noise, is 60 Hz. Therefore, the gain ratio of various resistance values at 60 Hz was compared (Figure 2b). Accordingly, a 50 GΩ resistor was adopted.

#### 2.1.3. Output Divider Feedback

Stray capacitance (*C_stray_*) represents the coupling capacitance value between the electrode and the guard ring. An increase in *C_stray_* results in a decrease in the total gain of the circuit. To reduce the effect of stray capacitance, the two ends of the stray capacitance must be maintained at similar voltages. By using the output divider feedback to transmit OPA output signals to the connection point of the input stray capacitance and the guard ring, the divider feedback signal can be distributed to reduce stray capacitance. Therefore, the divider feedback signal requires sufficient current drive force and a suitable divider ratio. The OPA of the electrode also needs a driving voltage to be maintained in the active region all the time for the high sensitivity of the sensing signal. Figure 3a depicts the effect of resistive divider ratio on gain; Figure 3b displays the effect of different *R_fl_* on gain given a fixed *R_fh_*. After analysis, the present study selected the parameters of *R_fl_* = 1 GΩ and *R_fh_* = 100 MΩ.

#### 2.1.4. Ground Guard Ring

The functions of the ground guard ring are to shield the electrode from external environmental interference—connecting a guard ring to a capacitance in series diverts the interference to ground potential, thereby increasing the SNR—and to provide an output feedback path to reduce stray capacitance.

### 2.2. Constructing the CECG Measurement System

The proposed noncontact ECG measurement system consists of a front-end electrode circuit, a CDRL feedback circuit, and a signal processing circuit (Figure 4). The two front-end electrodes are used to sense the ECG signals, the CDRL feedback circuit overcomes common-mode noise, and the signal processing circuit amplifies and filters signals. Sakuma et al. proposed the specifications of the CECG in general are that the cutoff frequency of high-pass filter (HPF) is less than 10 Hz and the cutoff frequency of low-pass filter (LPF) is 50 Hz [40]. Therefore, the proposed HPF was designed to have a corner frequency of 1 Hz, and the LPF was to have 42 Hz.

#### 2.2.1. CDRL

The CDRL circuit mainly eliminates common-mode noises (60 Hz) caused by coupling between the commercial energy line and the human body by transmitting the common-mode signal in the opposite direction. Thus, the circuit sends the signal toward the human body through the capacitive electrode, using the phase difference to inhibit common-mode noises. The capacitive value of C_sr_ is 1 pF, and the R, representing the input impedance of the OPA in CDRL circuit, is 10 KΩ. By using C_sr_ to increase the high-frequency signal gain and enabling inverted common-mode signals to pass through, the electrode reduces low-frequency signal gain and prevents the passing of physiological signals. Thus, the electrode offsets common-mode noise and enables the CECG measurement system to obtain a higher signal quality.

#### 2.2.2. Signal Processing Circuit

The signal processing involved amplification and filtering of the signals, which were achieved via the instrumentation amplifier circuit, high-pass filter, and low-pass filter. The circuit is pictured in Figure 5. Left and right electrode signals enter the signal processing circuit through the connection with signal inputs. First, the instrumentation amplifier circuit is used to obtain the CECG signal. This physiological signal, collected by the capacitive coupled electrode, is subsequently serially processed using a high-pass filter, a low-pass filter, and two amplifier circuits.

#### 2.2.3. Data Acquisition

The system employs an analog-to-digital converter and acquisition card (NI-DAQ-6009) to convert analog signals to digital signals. The acquisition card is a hot-swap I/O module with 8 analog input channels, 2 analog output channels, and 12 digital input and output channels. The card features a 32-bit counter channel for analog data collection. The measurement channels are separately connected to the output ends of the CECG signal processing circuit for data collection. Subsequently, LabVIEW NI-DAQmx software is used for real-time data acquisition; the data is then saved as a general digital data file. MATLAB is subsequently employed for CECG digital data analysis.

### 2.3. Experiments

To verify whether the capacitive electrodes effectively reduce noise interference from external and motion artifact interference, simulated and human experiments were performed. The proposed electrode was compared with classic capacitive electrodes, which have high input impedance and are the most commonly used CECG measurement method. The structure of the classic electrode is shown in Figure 6. The value of input resistor (*R_load_*) is 50 GΩ.

#### 2.3.1. Simulated Testing Experiment

(1)Simulated Testing System Development

Figure 7 presents the regular ECG electrode measurement method of IEC 60601-2-47 [41]. The present study employed two 11 cm × 13 cm acrylic boards to simulate nonconductive mediums (i.e., clothing) and glued conductive aluminum foil between the two acrylic boards to simulate the human skin. The signal from the signal generator (Agilent 33220A function/arbitrary generator) passed through a 1000-to-1 step-down circuit to simulate the human ECG signal. The amplitude of the simulated ECG signal at generator output is 1 Vp-p. Therefore, a signal attenuation of 60 dB is produced by the step-down circuit. This signal was then transmitted to the aluminum foil. Both electrodes were connected to the top acrylic board, representing the process of using electrodes to measure physiological signals (i.e., ECG signals) through clothing. The thickness of the non-conductive acrylic medium is 0.1 cm. The CDRL electrode was attached to the bottom acrylic board, as indicated in Figure 6. The left and right electrode signals first passed through the signal processing circuit, underwent analog-to-digital conversion through the DAQ card, and were displayed and recorded as ECG signals on the computer screen.

(2)Interference Simulation Experiment

A direct current power source was connected to a metal panel to provide the panel with electric potential, which simulated the electric potential of the human body under the voltage divider rule (1 V). The metal plate was moved forward and backward as well as left and right using a motor at different positions to simulate the effect of human-generated moving interference sources on the CECG measurement system. For a full description of the employed interference platform, please consult the previous study of the present research team [33].

#### 2.3.2. ECG Measurement Experiment

(1)ECG Measurement System Development

The ECG measurement system, presented in Figure 8, is largely similar to the simulated measurement system, with the main difference being that the capacitive electrode is directly fixed to the participant’s clothing to measure the ECG. The experimental process was as follows: First, the participant was seated and the CECG electrode was fixed against the chest with chest straps. Subsequently, other participants (representing interference sources) walked around the participant at various distances to create interference. The experiment was performed using the proposed active electrode and classic active electrodes, and the results were compared to validate the noise immunity capacity of the proposed active electrode.

(2)ECG Measurement Experiment Interference Methods

Four interference methods were employed in the human measurement experiment: the interferer walking forward and back in front of the participant, the interferer walking left and right in front of the participant, the interferer walking left and right at the sides of the participant, and the participant shaking their body forward, backward, left, and right. The first three interference methods simulated the interference caused by others walking in the vicinity of the participant; each method was separately performed at distances of 25, 50, and 75 cm. The fourth method simulated the motion artifact interference generated by the participant’s body movements.

## 3. Results

### 3.1. Simulated Measurement Results

#### 3.1.1. Signals Measured when Forward- and Backward-Moving Interference Was Positioned at the Front of the Electrode

To facilitate comparison, this represents the classic CECG wave and the measured CECG wave as Waveforms I and II, respectively. Interference was activated during 10–20 s. Figure 9 displays the signals measured when a forward- and backward-moving interference source moved at distances of 75–50, 50–25, and 25–0 cm in front of the electrode. According to the results of Waveform I, the interference increased as interference source distance decreased. Overall, the waveforms of Waveform II were more stable than those of Waveform I. When the interference source was at a distance of 25–0 cm, the signals of Waveform I were near saturation, whereas the baseline of Waveform II only trembled slightly (Figure 9c). Furthermore, Waveform II hardly sustained any interference when the distance of the interference source was 75–50 cm (Figure 9a).

#### 3.1.2. Signals Measured When Left- and Right-Moving Interference Was Positioned at the Front of the Electrode

Figure 10 depicts the signals measured when a left- and right-moving interference source was positioned 75, 50, and 25 cm in front of the electrode. The interference sustained by Waveform I was more substantial than that sustained by Waveform II (Figure 10a–c). However, the interference sustained by Waveform I in Figure 10 was dramatically less than that in Figure 9, indicating that Waveform I sustained more interference from the forward- and backward-moving interference than the left- and right-moving interference. Notably, neither of these interference considerably affected Waveform II.

#### 3.1.3. Signals Measured When Left- and Right-Moving Interference Was at the Sides of the Electrode

Figure 11 presents the signals measured when a left- and right-moving interference source was positioned at various distances at the sides of the electrode (75, 50, and 25 cm). The results of Waveform I concur to those of the previous two interference methods, with the waveform sustaining the most and least interference when the interference source was at distances of 25 (Figure 11c) and 75 cm (Figure 11a), respectively. Waveform II sustained minor interference when the interference source was at 25 cm (Figure 11c) but negligible interference at distances of 50 and 75 cm (Figure 11a,b).

#### 3.1.4. SNR under Interference at Different Distances

Figure 12 presents the SNR of the proposed electrode and the classic electrode when the forward- and backward-moving interference was positioned at various distances in front of the electrode. The SNR is calculated using Equation (4). P(Rpeak) is the power of the 100-ms interference-free ECGs centered on the detected R peak and $P\left(Rpeak_{Noise}\right)$ is the power of the 100-ms interference ECGs centered on the detected R peak. As expected, the SNR increased at greater distances, and the proposed electrode demonstrated more favorable performance than the classic electrode.
(4)$$\mathrm{SNR}=20\log\frac{P\left(Rpeak\right)}{P\left(Rpeak_{Noise}\right)}$$

### 3.2. Human Measurement Results

#### 3.2.1. Signals Measured with a Forward- and Backward-Moving Interferer in Front of the Participant

Figure 13 presents the signals measured with a forward- and backward-moving interferer in front of the participant at distances of 75–50, 50–25, and 25–0 cm. The results of Waveform I concur to those of the simulated interference experiment—interference increased as interference source distance decreased (Figure 13a–c). Additionally, the Waveform II appeared more stable than Waveform I, validating that the CECG signals measured using the proposed electrode sustained less noise interference.

#### 3.2.2. Signals Measured with a Left- and Right-Moving Interferer in Front of the Participant

Figure 14 displays the signals measured with a left- and right-moving interferer in front of the participant at distances of 75, 50, and 25 cm. The results indicated that Waveform I sustained some interference when the interferer was at distances of 50 and 25 cm (Figure 14b,c) and less interference when the interference source was at a distance of 75 cm (Figure 14a). When the interferer was moving at a distance of 25 cm, Waveform II exhibited a slight trembling, but the R-wave of the ECG was not disrupted (Figure 14c). When the interference distance was set to 50 and 75 cm, Waveform II did not sustain interference (Figure 14b,c).

#### 3.2.3. Signals Measured when a Left- and Right-Moving Interferer Was at the Sides of the Participant

Figure 15 illustrates the signals measured when a left- and right-moving interferer was 75, 50, and 25 cm to the sides of the participant. Waveform I sustained considerable interference in all three distances. By contrast, Waveform II demonstrated more stable performance (Figure 15a–c) and did not sustain interference when the interference distance was set to 75–50 cm.

#### 3.2.4. Signals Measured When Interference Was Caused by the Participant’s Body Movement

Figure 16 displays the signals measured when interference was generated by the motion artifact interference caused by the participant’s body movement. Figure 16a depicts the waveform measured when the participant moved his body forward and backward, and Figure 16b displays the waveform produced when the participant moved left and right. In the 10–20 s period, motion artifact interference was generated in both scenarios. The interference was stronger in Waveform I. Although motion artifact interference was also generated when measuring Waveform II, the peak of the R-wave in Waveform II remained visible.

#### 3.2.5. Comparison between ECG Signals Measured by Contact and Noncontact Electrodes

Figure 17 compares the ECG signal measured by contact and noncontact electrodes. The signal measurement duration was 60 s, with no interference during the first and last 15 s. During the 15–45 s, the interferer moved between 50 and 25 cm in front of the participant, similar to the interference method in Section 3.2.1. Figure 17 indicates substantial stray interference in the middle 30 s of signal measurement by the conventional CECG electrode, whereas the proposed CECG electrode did not sustain interference. A comparison between the 15–45 s period in Figure 17a,c revealed a 98.6% correlation in R–R interval, whereas that of Figure 17a,b was 56.4%.

## 4. Discussion

The present study proposed a noncontact CECG measurement system capable of stable ECG signal measurement. To overcome a major problem of current capacitive electrodes, namely susceptibility to interference, circuit parameter optimization analysis was performed to determine the most suitable circuit design framework. The proposed design can overcome the problems in current capacitive sensors, particularly those related to their susceptibility to environmental interference and sensor shaking, thereby successfully improving the quality of ECG signals measured by noncontact measurement systems. Noise interference has been a constant bottleneck in the commercialization of CECG measurement circuits. Various solutions for overcoming noise interference have been proposed [15,32,34,35,36]. However, few solutions have achieved stable CECG signal measurement in a physical environment. The present study simulated an environment with various interference to the measurement of CECG signals. The proposed CECG measurement system demonstrated outstanding antinoise capabilities to overcome environmental and motion artifact interference. This system may serve as a favorable design reference for the practical development of CECG sensors.

The capacitive electrode design process consisted of conducting circuit parameter property analysis, employing common-mode noise canceling, selecting electrode components, and increasing the gain ratio to design the capacitive electrode. Furthermore, a simulated interference testing experiment platform was developed to simulate external interference sources and the interference factors generated by human movement. The effect of interference on the CECG system was observed, and the interference generated by interference sources at various distances was compared to clarify the problems faced by existing CECG systems. The experiment results revealed that conventional capacitive electrodes sustained interference from interference sources or human movement at distances of ≤75 cm (Figure 9, Figure 10, Figure 11, Figure 12, Figure 13, Figure 14 and Figure 15). Furthermore, this interference noise increased as the interference distance decreased. The proposed CECG is capable of withstanding most noise interference. Although minor interference was observed when the interference source was positioned at ≤25 cm, the R-wave peak value of the measured ECG signal remained visible. Moreover, the proposed CECG electrode is capable of inhibiting motion artifact interference generated by electrode movement (Figure 16). Thus, the proposed capacitive electrode is more suitable for application in commercial and portable products.

A comparison between the ECGs measured using the conventional ECG electrode and the CECG electrode in Figure 17 revealed a 98.6% correlation in the R–R intervals. The two ECGs were largely similar, indicating that the accuracy of the proposed CECG measurement system is close to that of conventional contact ECG electrodes. Therefore, the current authors argue that the proposed CECG measurement technology may replace conventional contact ECG electrodes in long ECG recordings.

In terms of circuit parameter selection, researchers should pay greater attention to selecting suitable components. Figure 2 indicates that a higher input resistance (*R_load_*) resulted in a higher gain. However, this study did not select a 100-GΩ resistor because an excessively large resistance may result in the output of motion artifact interference, which, of course, hampers the inhibition of motion artifact interference. Therefore, tradeoffs are necessary in the selection of input resistors. Operators must ensure that the ECG signal can be acquired and consider the effect of the resistor on motion artifact interference.

The present study compiled design combinations with a high gain ratio to overcome the problem of interference susceptibility in CECG. In the electrode design process, manufacturers should endeavor to design near-identical electrodes for CECG measurement to prevent common-mode conversional loss. Moreover, because CECG is susceptible to external interference, environmental interference factors should first be eliminated when measuring ECG to prevent their effect on the measured data. For example, when performing ECG measurements, the operating condition of surrounding instruments and devices should remain constant throughout the process to ensure that human movement is the only interference source, thus improving the reproducibility of the interference and result data.

## 5. Conclusions

The present study proposed a novel capacitive electrode and devised a CECG measurement system based on said electrode. In addition to performing in-depth research on the effect of noise, this study provides solutions to noise filtering and signal processing. Furthermore, a simulated test platform for assessing the antinoise properties of the CECG measurement system was developed. Empirical results of simulated and actual ECG measuring experiments revealed that the proposed CECG electrode outperformed conventional CECG electrodes. The proposed CECG electrode may serve as a valuable reference for future practical applications of noncontact ECG measurement systems.

## Figures and Tables

**Figure 1 sensors-21-03668-f001:**
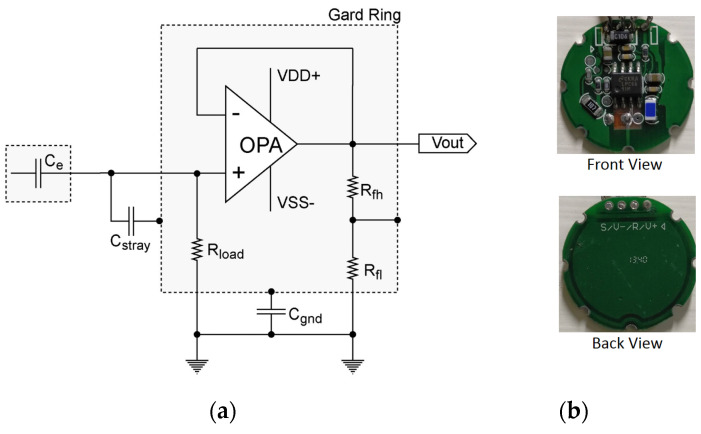
Active capacitive electrode design. (**a**) Active electrode equivalent circuit and (**b**) pictures of the active capacitive electrode.

**Figure 2 sensors-21-03668-f002:**
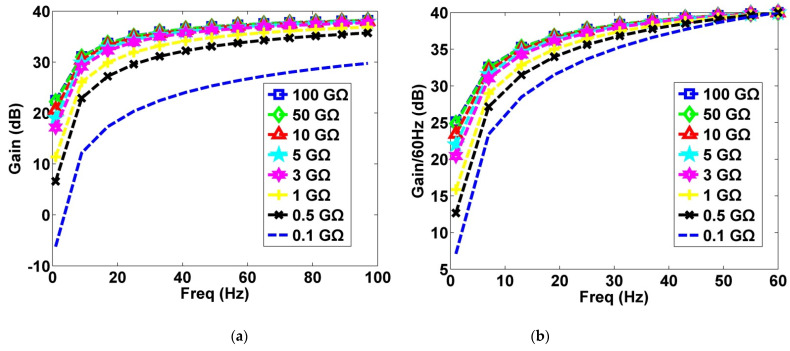
Effect of different resistance values (*R_load_*) on gain (V_out_/V_i_). (**a**) Effect of resistance values between 100 MΩ and 100 GΩ on gain; (**b**) effect of resistance values between 100 MΩ and 100 GΩ on gain ratio at 60 Hz.

**Figure 3 sensors-21-03668-f003:**
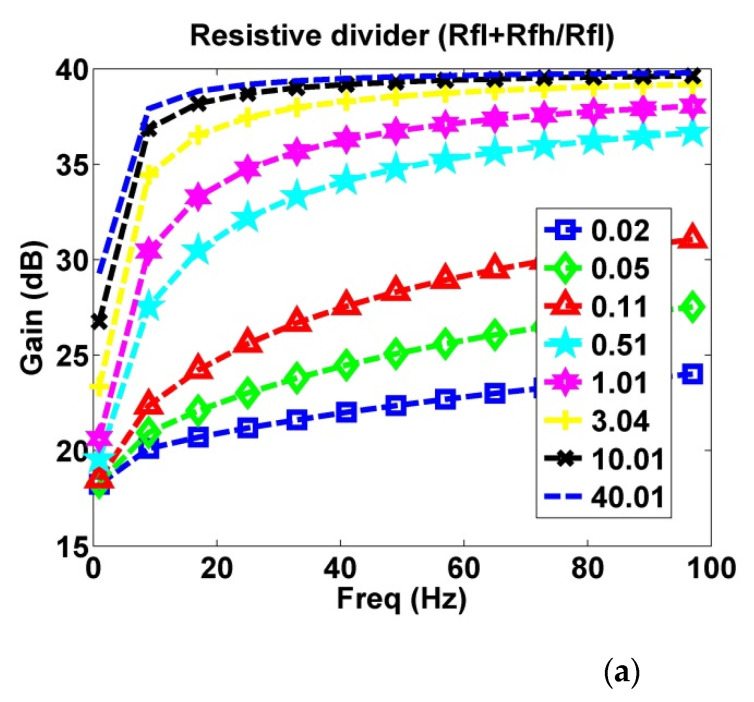
Effect of output divider feedback on gain. (**a**) Effect of a 0.02–40.01 (*R_fl_* + *R_fh_*)/*R_fl_* ratio on gain; (**b**) effect of 1 MΩ to 1 GΩ *R_fl_* values on gain.

**Figure 4 sensors-21-03668-f004:**
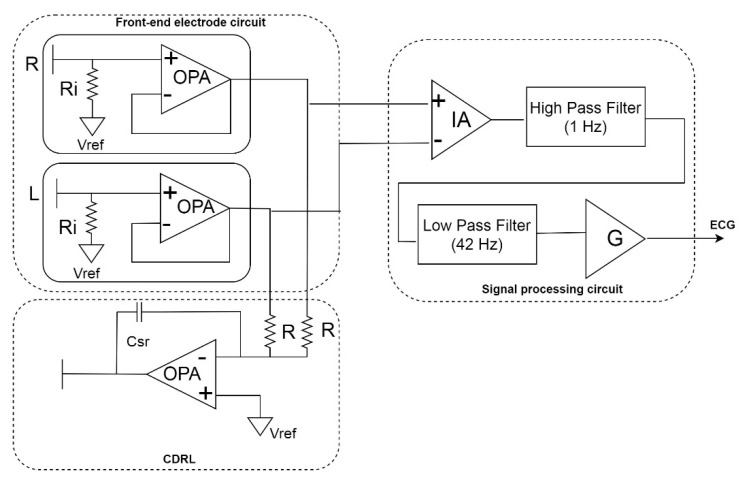
Design of the noncontact ECG measurement circuit.

**Figure 5 sensors-21-03668-f005:**
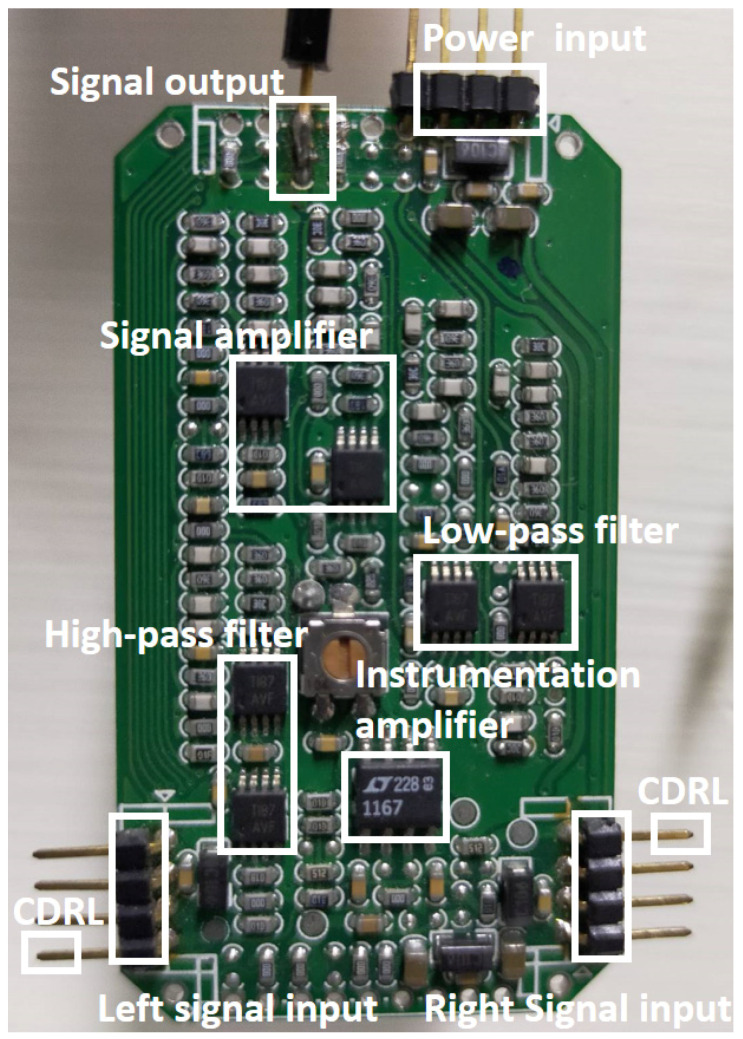
Back-end signal processing and amplifier circuit.

**Figure 6 sensors-21-03668-f006:**
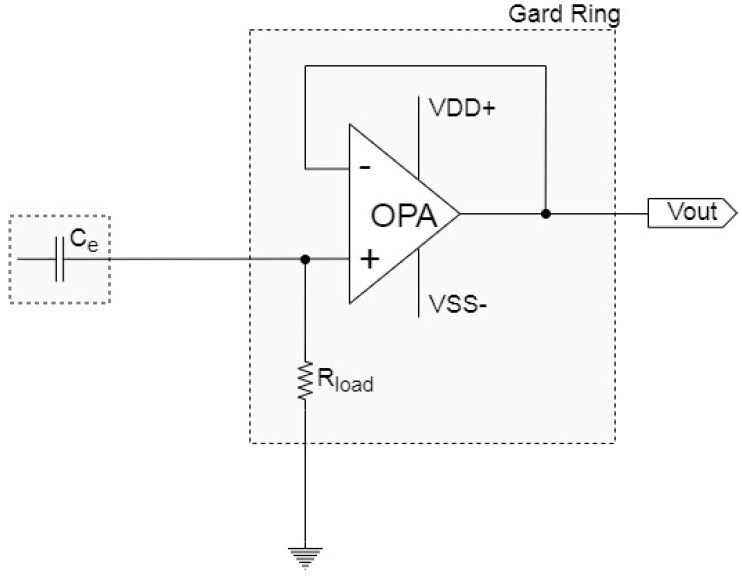
Classic electrode equivalent circuit.

**Figure 7 sensors-21-03668-f007:**
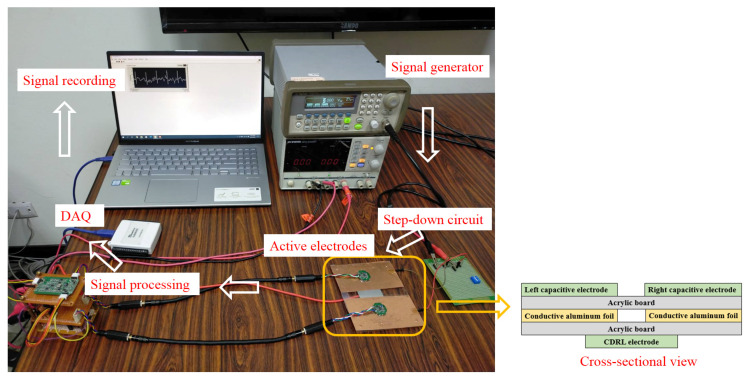
Simulated measurement system.

**Figure 8 sensors-21-03668-f008:**
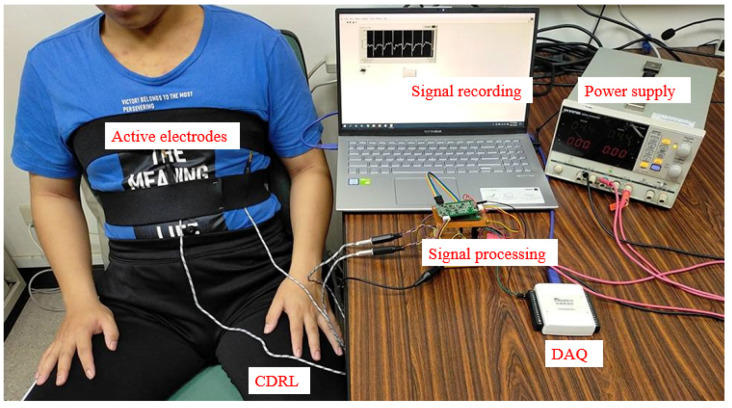
Human measurement system development.

**Figure 9 sensors-21-03668-f009:**
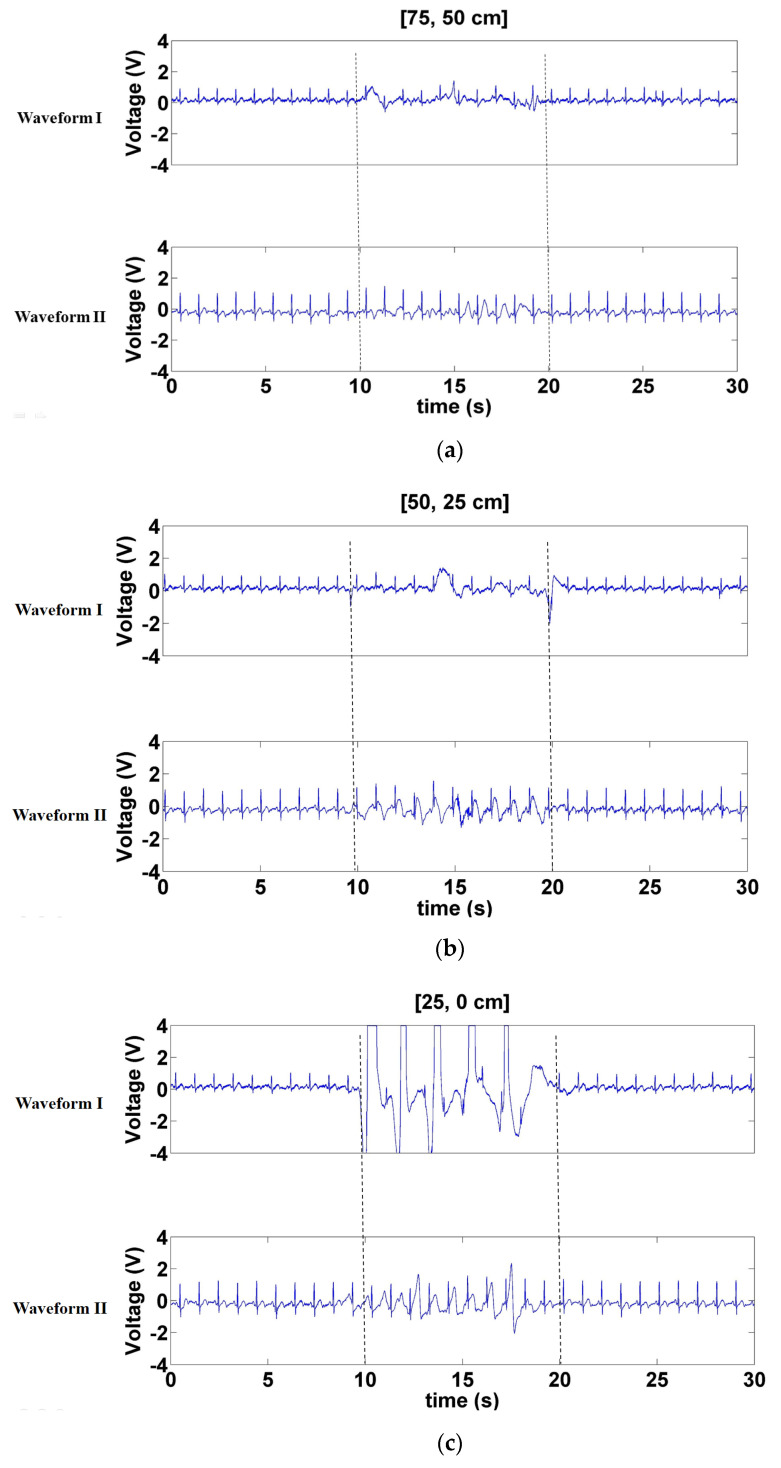
Signals measured when the forward- and backward-moving interference moved at distances of (**a**) 70–50 cm, (**b**) 50–25 cm, and (**c**) 25–0 cm in front of the electrode.

**Figure 10 sensors-21-03668-f010:**
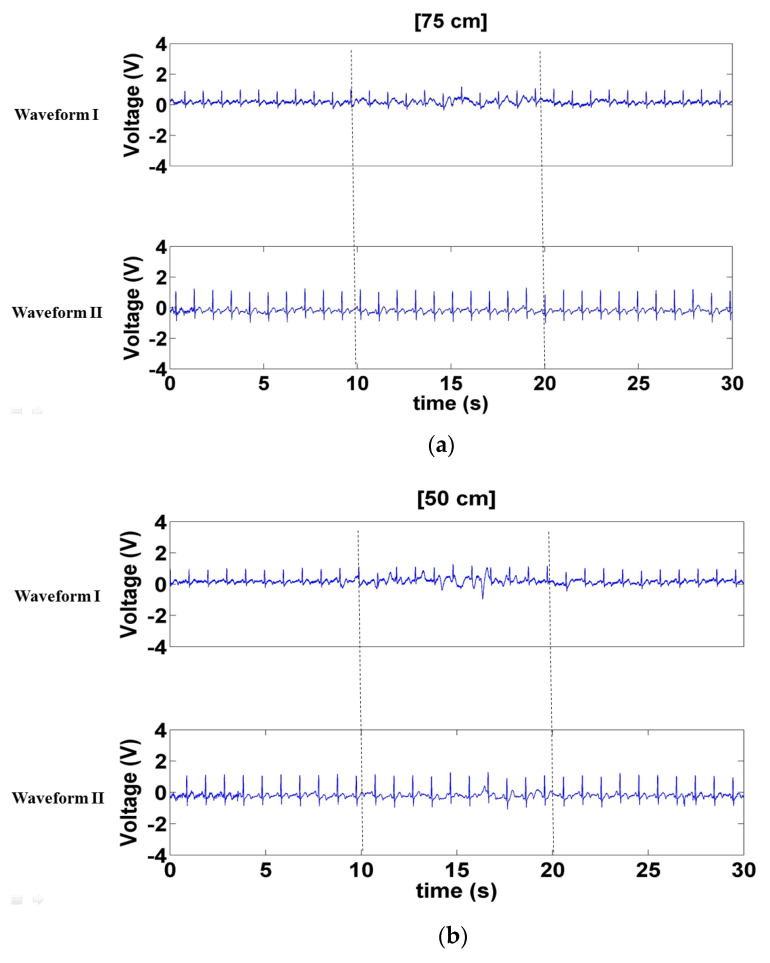
Signals measured when left- and right-moving interference was positioned (**a**) 75 cm, (**b**) 50 cm, and (**c**) 25 cm in front of the electrode.

**Figure 11 sensors-21-03668-f011:**
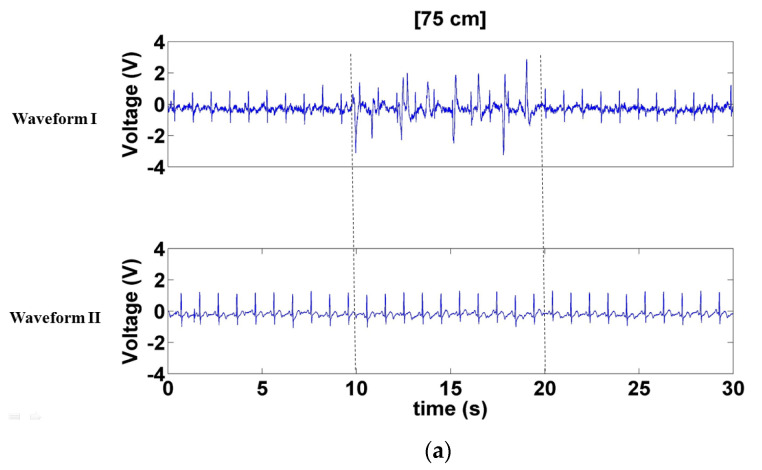
Signals measured when the interferer moved left and right at distances of. (**a**) 75 cm, (**b**) 50 cm, and (**c**) 25 cm at the sides of the electrode.

**Figure 12 sensors-21-03668-f012:**
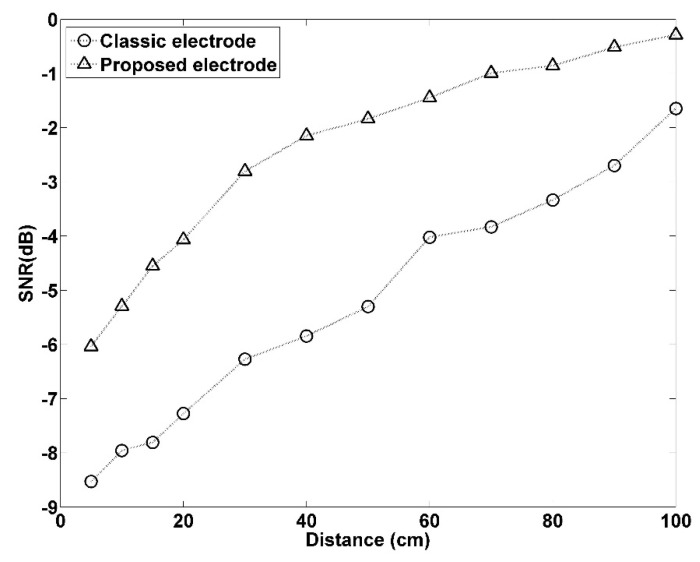
Signal-to-noise ratio (SNR) comparison of interference at different distances.

**Figure 13 sensors-21-03668-f013:**
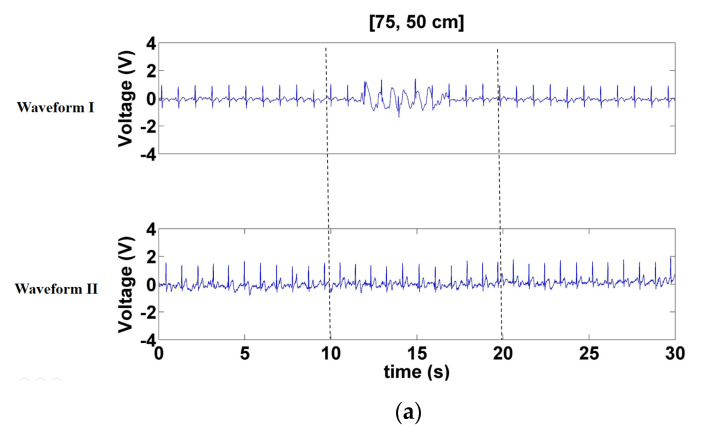
Signals measured when forward- and backward-moving interferers moved at distances of (**a**) 75–50, (**b**) 50–25, and (**c**) 25–0 cm in front of the participant.

**Figure 14 sensors-21-03668-f014:**
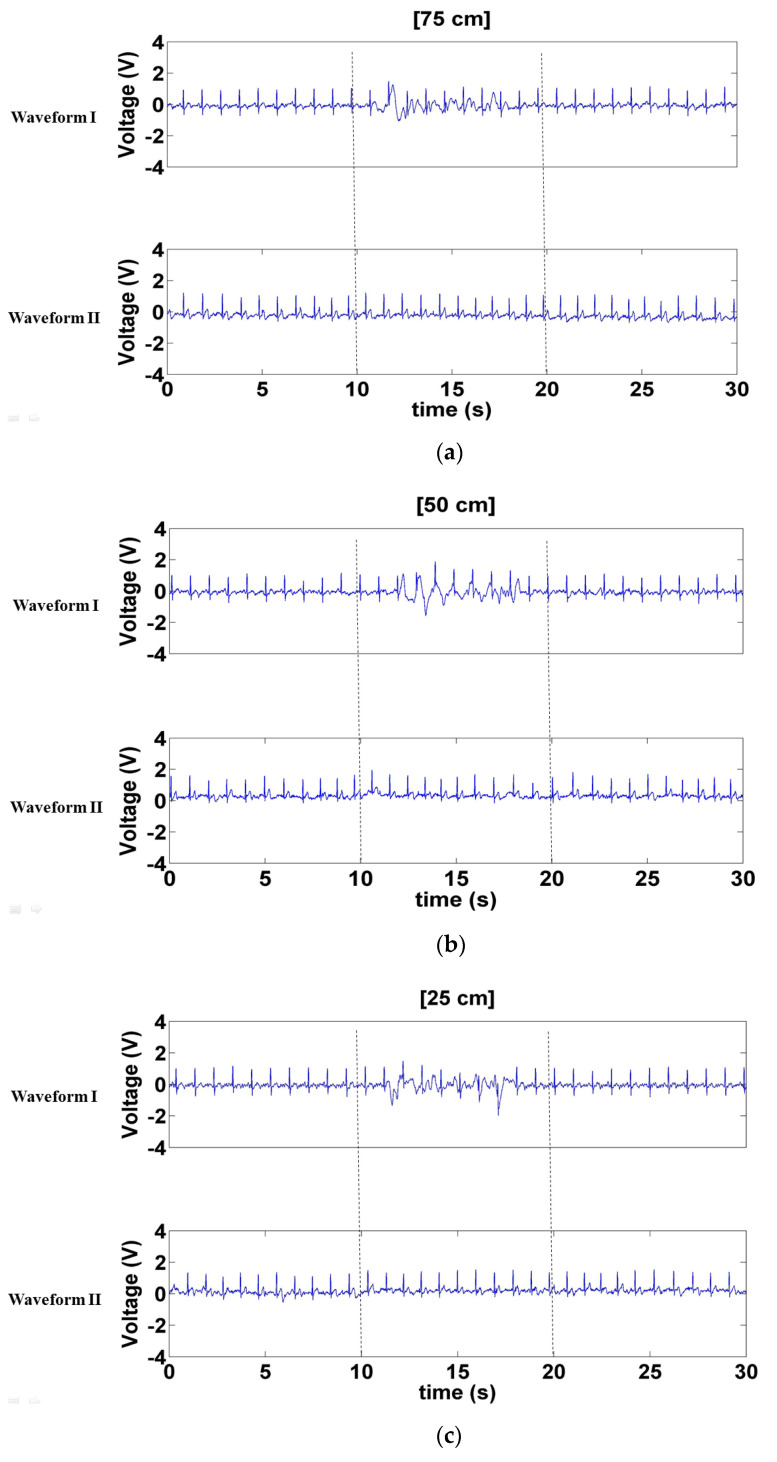
Signals measured when a left- and right-moving interferers were positioned (**a**) 75, (**b**) 50, and (**c**) 25 cm in front of the participant.

**Figure 15 sensors-21-03668-f015:**
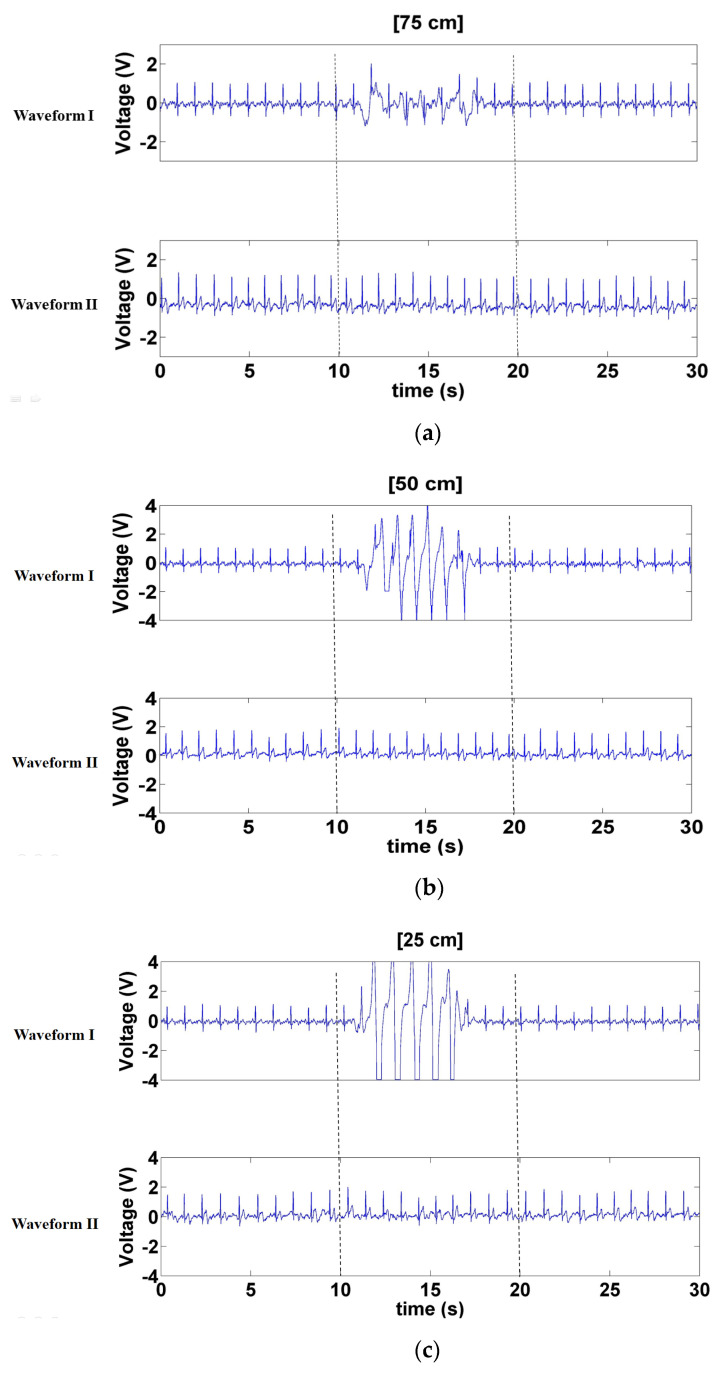
Signals measured when a left- and right-moving interferer was (**a**) 75, (**b**) 50, and (**c**) 25 cm to the sides of the participant.

**Figure 16 sensors-21-03668-f016:**
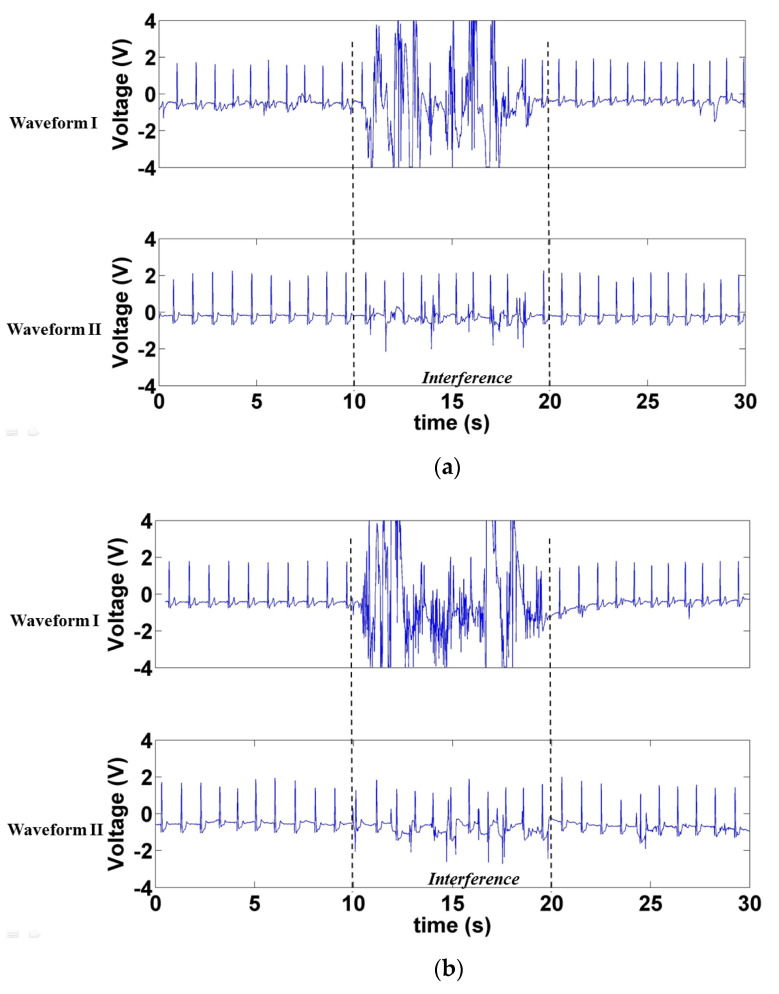
Signals measured under motion artifact interference generated by the participant’s (**a**) forward and backward and (**b**) left and right body movements.

**Figure 17 sensors-21-03668-f017:**
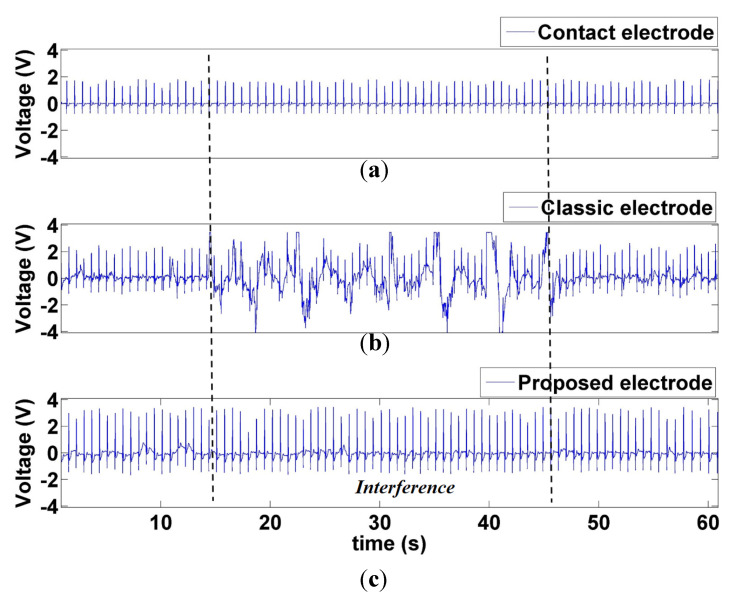
Comparison between the interference at 15–45 s during ECG signal measurement by the (**a**) conventional contact electrode, (**b**) proposed CECG electrode, and (**c**) classic CECG electrode.

## Data Availability

Not applicable.

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
