# Peer review of "Novel Stable Capacitive Electrocardiogram Measurement System"

_sensors, 2021, doi:10.3390/s21113668_

Round 1

Reviewer 1 Report

The paper deals with designing a capacitive ECG electrode that is stable against motion artefacts and environmental interference. Although the results are interesting and seem promising, the paper has some serious flaws that should be solved before publishing. I have the following comments on the paper:

  • Figure 1: Labels of capacitors and resistors use a small and unreadable font.
  • Line 112: “A resistance value of >1 GΩ is unsuitable for ECG measurement”. I think there should be < 1 GΩ.
  • Line 115: Should be Figure 2b instead of Figure 1b.
  • Figure 2: What are the other parameters (value of Ce, Cstray, Cgnd) used for the graphs? Why the gain use units of percentage? Please consider using typical decibel units—the same for Figure 3.
  • Figure 4: Why the cut-off frequency of the high pass filter is 1 Hz? The main bandwidth of ECG is 0.5 – 35 Hz (Line 113).
  • Figure 5: Where are located pins for connection of DRL electrode?
  • Line 180 – 181: “The proposed electrode was compared with classic capacitive electrodes, which have high input impedance and are the most commonly used CECG measurement method [33].” The “classic electrodes” which were used in experiments have to be described in detail. I think you don’t use the same electrode construction described in [33] because it is very specific. You should provide the schematic and construction of the “classic electrode” to make the results in Figures 8 – 16 more trustable.
  • In the simulated testing experiment, the thickness of the non-conductive medium is not specified. What was the amplitude of the ECG signal at generator output?
  • The figures depicting measured signals in six graphs should be resized to full page width because the signals are significantly compressed.
  • The ECG signals (waveform I) in Figures 8 and 9 have small negative values of QRS complex compared to waveform II. Please explain.
  • The positive values of the most ECG signals (Figure 10, 11, 12, 13, etc.) seems to be saturated because all R waves have the same magnitude what is impossible (signals in Figure 16 seems to be correct). Please explain.
  • Figure 11: How the SNR was calculated? Negative SNR means that signal power is lower than noise power.  Is it suitable for signals in Figure 10?
  • Line 332 – 333: I think there is a mistake in “…in Figure 16a,b revealed a 98.6% correlation in R–R interval, whereas that of 16a,c was 56.4%.”

Reviewer 2 Report

I suggest 2 faults to be corrected:

  1. Work with capacitance electrodes was performed in the 1970's and some pertainent references should be included from early work
  2. It should be mentioned that capacitance electrodes require a driving voltage
  3. How do the current needs of the capacitance affect battery life of the recording system?  How does that compoare to regular electrodes?

Round 2

Reviewer 1 Report

I thank the authors for the corrections. All of my comments have been addressed. I recommend publishing the paper in the present form.